# Establishment of a Highly Efficient In Vitro Propagation System of *Diospyros lotus*

**Yang Liu** [1,2,3], **Xiaoyu Lu** [1,2,3], **Hui Zhang** [1,2,3], **Shuzhan Li** [1,2,3] and **Ze Li** [1,2,3],*

1. Key Laboratory of Cultivation and Protection for Non-Wood Forest Trees, Ministry of Education, Central South University of Forestry and Technology, Changsha 410004, China
2. The Belt and Road International Union Research Center for Tropical Arid Non-Wood Forest in Hunan Province, Changsha 410004, China
3. The Key Laboratory of Non-Wood Forest Products of State Forestry Administration, Central South University of Forestry and Technology, Changsha 410004, China

* Correspondence: lize1853@163.com; Tel.: +86-856-23416

**Abstract:** Persimmon (*Diospyros*) is an economically important tree widely cultivated for woody grain production in China, and *Diospyros lotus* is mainly used as the grafting stock of persimmon. However, the breeding of stress-resistant rootstocks of *D. lotus* using molecular means has yet to be achieved; in particular, an efficient blade-regeneration system has not been perfected to date. This study examined the effects of different plant-growth regulators and concentrations on the primary culture of stems with buds, the induction of leaf callus, the differentiation of adventitious shoots, and rooting culture of *D. lotus*. The optimal formula for inducing axillary buds from stems with buds was 1/2 Murashige and Skoog (MS) medium containing 2.0 mg/L 6-benzylaminopurine (6-BA) and 0.5 mg/L naphthaleneacetic acid (NAA), in which the induction rate of axillary buds approached 67.1%. The best medium formula for leaf callus induction was 1/2 MS medium containing 2.0 mg/L 6-BA and 0.5 mg/L NAA. Then callus was transferred to 1/2 MS medium containing 2.0 mg/L 6-($\gamma$,$\gamma$-dimethylallylamino)purine (2iP), 2.0 mg/L thidizuron (TDZ), and 40 g/L sucrose to induce adventitious shoots after dark culture for 48 h, resulting in 7.9 shoots per explant and a 75.2% induction frequency of adventitious shoots. In addition, it was difficult to induce adventitious shoots from callus after six times of continuous transfer and differentiation. The adventitious shoots were transferred to 1/2 MS medium containing 2.0 mg/L zeatin (ZT) and 2.0 mg/L 2iP for proliferation culture, in which the multiplication coefficient approached 7.6. The adventitious shoots after multiplication were inoculated into 1/2 MS + 1.0 mg/L IBA + 0.5 mg/L NAA medium, the rooting rate was 70.2%, and the average number of heels was 9.6. Thus, studies in this area are expected to facilitate rapid and excellent growth, as well as theoretical support for factory saplings' care and molecular breeding.

**Keywords:** adventitious shoots; take root; *Diospyros lotus*; disinfection; tissue culture





## 1. Introduction

*Diospyros lotus* L. belongs to the Ebenaceae family of woody angiosperms, which is widely distributed in China [1]. Due to its strong resistance to drought and high grafting affinity, it is the primary rootstock for grafting persimmon trees [2]. Its fruit has high medicinal value, effective for the treatment of kidney stones, cancer, and bacterial infections [3,4]. Research on *D. lotus* has recently concentrated on phylogenetic analysis, molecular markers, resistance, nutritional makeup, and medicinal composition [5].

The most common way to reproduce *D. lotus* is through seed, although this method frequently produces offspring with unstable genetics [6]. Breeding using genetic transformation not only shortens the culture cycle but also stabilizes the genotype. An efficient tissue culture system is an important prerequisite for genetic transformation. Recent studies on persimmon tissue culture and genetic transformation have focused on Japanese

persimmon (*Diospyros kaki* Thunb), while relatively little research has been performed on its rootstock [5]. The time required to breed new plant types has been reduced significantly, due, in part, to advances in molecular breeding technology. Applying this technology to create new varieties of *D. lotus* seeds will provide excellent stress-resistant stock for the sweet persimmon industry [7].

There have been numerous studies on persimmon tissue culture; however, issues related to persimmon growth, including browning of explants, weak growth of adventitious shoots induced by callus, and difficulties in rooting have not been completely solved [8]. In addition, there are other challenges with the genetic transformation process of *D. lotus*, such as a high browning rate of explants, a low regeneration rate of adventitious shoots, the ease with which false-positive saplings might develop, and difficult rooting of positive plants [9]. In the case of persimmon genetic transformation, a poor transformation rate, difficulties in acquiring the positive shoots, and the inability to alter genes are particularly pronounced [5]. The establishment of a complete tissue culture system can effectively provide ideas for these problems.

There are two main reasons for choosing *D. lotus* as the material in this study. First, it is a diploid plant with relatively few alleles, this makes the later genetic transformation more effective. Second, its whole genome sequence has been published, which makes it easier to sequence and verify after genetic transformation; thus, an efficient tissue culture system can be widely used in subsequent genetic transformation [10]. Investigations into tissue culture of *D. lotus* not only provide a scientific basis for the study of functional genes of its fruit development but also serve as a technical guide for the study of adaptive genetic improvements in its rootstock and as a point of reference for similar studies involving other woody plants [11]. Thus, studies in this area are expected to facilitate rapid and excellent growth, as well as theoretical support for factory sapling care and molecular breeding.

## 2. Materials and Methods

### 2.1. Plant Materials

Three-year-old potted *D. lotus* saplings were grown in plastic containers (diameter 30 cm and height 35 cm) containing potting soil (mixed with peat, perlite, and vermiculite): loess (1:1, *v/v*) under outdoor conditions at Central South University of Forestry and Technology. Semi-lignified stems with buds were cut from potted saplings in May every year and cut into 3–4 cm long stems with single axillary buds for use.

### 2.2. Drug Information

This part contains all the drug companies and product numbers used in the study, which can achieve the same tissue culture effect when used by others later. The main drug information is as follows: 1/2 Murashige and Skoog (MS): Solarbio, M8526; (1/2N) MS: Shanghai Earthquake Organism, HZ1081-50L; MS: Solarbio, M8520; WPM: Solarbio, LA6881; Agar: Solarbio, A8190; sucrose: Hushi, 10021418; zeatin (ZT): Yuanye, S18003; thidizuron (TDZ): Solarbio, T8050; 6-($\gamma,\gamma$-dimethylallylamino)purine (2iP): Solarbio, I8330; naphthaleneacetic acid (NAA): Solarbio, N8010; indole acetic acid (IAA): Solarbio, I8020; kinetin (KT): Solarbio, K8011.

### 2.3. Sterilization of Explants

The leaves from the collected stems with buds were removed and washed two to three times with washing powder, then rinsed under running water for 30–40 min, put into a beaker, and disinfected on an ultra-clean bench using one of the four treatments given in the following for a specific amount of time. Each treatment and specific treatment time allotted included a group of 25 samples. All samples underwent disinfection with 75% ethanol for 25 s and then rinsing with sterile water three to five times, followed by one of the following treatments: continuous disinfection with 5% NaClO for 8, 10, and 12 min; immersion in 10% $H_2O_2$ solution for 8, 16, and 24 min; soaking in 2% NaClO solution for

15, 20, and 25 min; or shaking while soaking for 20, 25, and 30 min in a solution containing 0.1% $HgCl_2$. The survival, contamination, and browning rates were recorded after 25 days.

### 2.4. Induction and Culture of Axillary Buds

Semi-lignified stems with axillary buds were cultured on 1/2 Murashige and Skoog (MS), (1/2N) MS, MS, and woody plant medium (WPM) basal media with plant growth regulators of 6-benzylaminopurine (6-BA) (0, 1.0, 2.0, and 3.0 mg/L) and naphthaleneacetic acid (NAA) (0, 0.5, and 1.0 mg/L). The sprouting rate of axillary buds and the timing were evaluated after 30 days.

### 2.5. Leaf Disc Callus Induction

Leaves were removed from axillary buds, cut into $1 \times 1$ cm$^2$ large cubes, and cultured on 1/2 MS, (1/2N) MS, MS, and WPM media for callus induction. The medium for callus induction contained 6-BA (1.0, 2.0, and 3.0 mg/L) and NAA (0, 0.5, and 1.0 mg/L). Callus formation rate and browning rate were evaluated after 25 days.

### 2.6. Induction of Adventitious Shoots from Callus

The browning layer on the outermost callus induced by leaves was removed. Samples of $5 \times 5 \times 5$ mm$^3$ callus were cut and cultured on 1/2 MS, MS, WPM, and (1/2N) MS media containing different concentrations of thidizuron (TDZ) (1.0, 2.0, and 3.0 mg/L), 6-($\gamma,\gamma$-dimethylallylamino)purine (2iP) (1.0, 2.0, and 3.0 mg/L), and 0.5 mg/L NAA. Sucrose (30, 40, and 50 g/L) was added to the medium, and the adventitious shoot induction rate and induction coefficient were evaluated after 25 days of culture. Adventitious shoots coefficient = total number of callus-induced intimating buds/callus index

### 2.7. Observation of Callus Tissue Morphology

The browning layer on the outermost callus induced by leaves was removed, and $5 \times 5 \times 5$ mm$^3$ large callus samples were cut and cultured on 1/2 MS + 2.0 mg/L TDZ + 2.0 mg/L 2iP + 0.5 mg/L NAA medium. The induction coefficient and inductivity were evaluated for the first, second, third, fourth, and fifth generations of adventitious shoot induction. The growth status of callus cells was observed using paraffin sections. Paraffin sections were stained with hematoxylin. Induction coefficient = callus-induced number of adventitious shoots/total callus number; inductivity = callus index/total callus number of induced adventitious shoots.

### 2.8. Adventitious Shoot Proliferation in Culture

Adventitious shoots of about 2 cm were inoculated in medium containing zeatin (ZT) (1.5, 2.0, and 2.5 mg/L), 2iP (1.0, 2.0, and 3.0 mg/L), and indole-3-acetic acid (IAA) (0.05 mg/L). After 25 days of incubation, the growth and average plant height of the group saplings were counted. Multiplication coefficient = increase the index/the original number of individuals.

### 2.9. Rooting Culture

Adventitious shoots of about 2.5 cm were inoculated in medium containing indole butyric acid (IBA) (0.5, 1.0, and 1.5 mg/L) and NAA (0, 0.5, and 1.0 mg/L). Sucrose (15, 20, and 25 g/L) was added to the medium, the rooting rate and adventitious root number were evaluated after 40 days of culture. Take root rate = number of adventitious shoots rooting / number of adventitious shoots inoculated into the culture medium.

### 2.10. Conditions of the Culture

The basal medium contained 30 g/L sucrose, 7 g/L agar, and was pH 5.5–5.8. The culture temperature was $28 \pm 1$ °C, the light intensity was 50–60 μmol/m$^2$/s$^1$, the light time was 14 h/day, and the medium was sterilized in a 121 °C autoclave cooker for 20 min.

### 2.11. Statistical Analysis

All experiments were carried out as a completely randomized design with three replicates, and each treatment contained 90 explants, except for the genetic transformation process. The data were analyzed using one-way analysis of variance, followed by Duncan's multiple range test. The significance level was set at $p < 0.05$. Data are expressed as mean $\pm$ standard error (SE). All statistical analyses were carried out using the SPSS statistical software package version 22.0 (IBM Corp., Armonk, NY, USA).

## 3. Results

### 3.1. Sterilization of Explants

Disinfection of explants is a key step in tissue culture, proper disinfection methods can improve the survival rate of primary materials, reduce the browning rate, and obtain as many sterile materials as possible from the limited materials at the initial stage. Table 1 shows the browning and contamination rates of the stems. The buds disinfected with 75% ethanol for 25 s and in 0.1% $HgCl_2$ for 25 min were fewer in number, and the survival rates were significantly higher, than in those using other disinfectant treatments ($p < 0.05$). The survival rate of the stems with the buds sterilized in 0.1% $HgCl_2$ for 25 min was 25.0%, 27.6%, and 34.0% higher than the treatments of 2% NaClO for 15 min, 5% NaClO for 8 min, and 10% $H_2O_2$ for 16 min, respectively ($p < 0.05$). A comparison of the axillary bud induction rate, contamination rate, and browning rate indicated that the optimal method for stem sterilization of *D. lotus* buds was using 75% ethanol for 25 s, followed by 0.1% $HgCl_2$ for 25 min.

**Table 1.** Effects of disinfectant types and times on disinfection of stem with bud of *D. lotus*.

| 75% Ethanol | Disinfectant | Disinfection Time | Browning Rate % | Contamination Rate % | Axillary Bud Survival Rate % |
|---|---|---|---|---|---|
| 25 sec | 2% NaClO | 15 min | 30.8 ± 3.6d (33) | 36.7 ± 5.3b (33) | 25.0 ± 1.6b (23) |
| 25 sec | | 20 min | 45 ± 1.2c (41) | 33.3 ± 1.3b (30) | 21.7 ± 4.2b (20) |
| 25 sec | | 25 min | 51.7 ± 4.4b (47) | 30.0 ± 3.2bc (27) | 18.3 ± 3.6c (16) |
| 25 sec | 5% NaClO | 8 min | 36.7 ± 2.1d (33) | 36.6 ± 5.3b (33) | 26.7 ± 4.7b (24) |
| 25 sec | | 10 min | 43.4 ± 4.8c (40) | 35.0 ± 2.2b (32) | 21.6 ± 5.8b (19) |
| 25 sec | | 12 min | 65.0 ± 4.2a (59) | 16.7 ± 3.3d (15) | 18.3 ± 6.3c (16) |
| 25 sec | 0.1% $HgCl_2$ | 20 min | 33.7 ± 2.5e (12) | 21.3 ± 7.5c (19) | 45.0 ± 7.3a (41) |
| 25 sec | | 25 min | 35.0 ± 1.9e (13) | 11.7 ± 2.5d (11) | 53.3 ± 6.6a (48) |
| 25 sec | | 30 min | 42.1 ± 4.5d (28) | 11.2 ± 3.2e (10) | 46.7 ± 9.5a (42) |
| 25 sec | 10% $H_2O_2$ | 8 min | 38.4 ± 3.2d (35) | 48.3 ± 2.4a (43) | 13.3 ± 3.3c (12) |
| 25 sec | | 16 min | 39.9 ± 5.4c (41) | 40.8 ± 5.6a (37) | 19.3 ± 2.3c (17) |
| 25 sec | | 24 min | 64.4 ± 8.8a (58) | 23.3 ± 3.7c (21) | 11.7 ± 2.4c (11) |

Means followed by the same letters in rows are not significantly different at $p \leq 0.05$. The values in brackets are the real statistical data with the decimal point removed.

### 3.2. Induction and Culture of Axillary Buds

Germination of sterilized explants is difficult, choosing the appropriate medium can effectively improve the germination rate of explants. We inoculated the aseptic stem segments of *D. lotus* buds, sterilized using 75% ethanol for 25 s followed by 0.1% $HgCl_2$ for 25 min, into culture medium to which different plant growth regulator numbers had been added (Table 2). The axillary buds began to sprout after approximately 7 days (Figure 1a). After another 8 days of culture, the axillary buds began to grow leaves (Figure 1b). After 15 days of continuous culture, the height of new buds sprouted from the axillary buds approached 3.0 cm (Figure 1c). The aseptic stem segments of *D. lotus* germinated with different concentrations of plant growth regulators and basal media; however, the axillary bud sprouting rate varied greatly, as shown in Table 2. For the medium without plant growth regulators, the axillary bud sprouting rate was only 3.2% and the time required for axillary bud sprouting was 29.3 days. The worst results were obtained in the culture of stem segments with buds. When the NAA concentration was 0.5 mg/L, axillary bud sprouting increased at first and then decreased

with increasing 6-BA concentration. When the 6-BA concentration was 2.0 mg/L, the axillary bud sprouting rate, which was 67.1% ($p < 0.05$), was significantly higher than those of other media. The overall sprouting rate was lower with increasing 6-BA concentration at NAA concentrations of 1.0 mg/L, and there was no significant difference. By comparison, the best medium for axillary bud sprouting of *D. lotus* stem segments was 1/2 MS medium supplemented with 2.0 mg/L 6-BA and 0.5 mg/L NAA.

**Table 2.** Effects of different plant growth regulators and culture media on the primary culture of *D. lotus* stems with buds.

| Culture Medium | 6-BA mg/L | NAA mg/L | Sprouting Rate % | Sprouting Time D |
|---|---|---|---|---|
| 1/2 MS | 0 | 0 | 3.2 ± 0.2e (3) | 29.3 ± 3.4 a |
| 1/2 MS | 1 | 0 | 27.3 ± 5.7d (25) | 22.0 ± 2.7 b |
| 1/2 MS | 2 | 0 | 30.4 ± 1.3d (27) | 20.4 ± 5.9 bc |
| 1/2 MS | 3 | 0 | 25.2 ± 6.9d (23) | 21.8 ± 5.4 b |
| 1/2 MS | 1 | 0.5 | 42.4 ± 2.2c (38) | 19.6 ± 3.3 c |
| 1/2 MS | 2 | 0.5 | 67.1 ± 4.7a (60) | 17.2 ± 1.9 c |
| 1/2 MS | 3 | 0.5 | 53.3 ± 6.6b (48) | 21.4 ± 4.7 b |
| 1/2 MS | 1 | 1 | 22.7 ± 3.6d (20) | 23.7 ± 5.8 b |
| 1/2 MS | 2 | 1 | 25.8 ± 8.2d (23) | 23.0 ± 3.6 b |
| 1/2 MS | 3 | 1 | 24.2 ± 3.3d (22) | 26.1 ± 2.7 a |
| (1/2N) MS | 2 | 0.5 | 55.6 ± 3.8b (50) | 22.5 ± 2.3 b |
| MS | 2 | 0.5 | 40.2 ± 8.2c (36) | 23.6 ± 1.9 b |
| WPM | 2 | 0.5 | 43.5 ± 6.1c (39) | 20.4 ± 4.7 bc |

Means followed by the same letters in rows are not significantly different at $p \leq 0.05$. The values in brackets are the real statistical data with the decimal point removed.

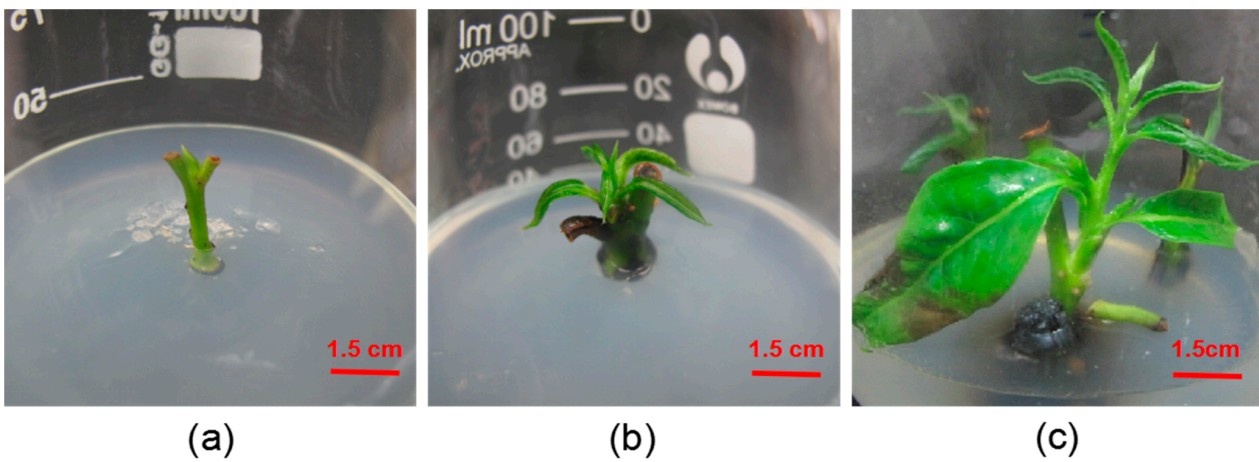

(a)  (b)  (c)

**Figure 1.** Growth status of the axillary buds of *D. lotus*. (**a**) Stem with buds inoculated in culture medium after 7 days. (**b**) Stem with buds grows leaves after 15 days. (**c**) Stem with buds growing to 3 cm in 25 days.

*3.3. Leaf-Disc Callus Induction*

Callus induction is a critical step in the process of leaf regeneration and selecting the appropriate medium can greatly shorten the leaf discs' regeneration time. We inoculated leaf discs of *D. lotus* into culture medium, to which different amounts of plant growth regulators were added (Table 3). No browning appeared in the medium of leaf-induced calli at 5 days (Figure 2a). Figure 2b shows the calli after 10 days. The callus dimensions were $1.5 \times 1.5$ cm$^2$ after 25 days (Figure 2c). When the 6-BA concentration was constant, callus induction increased at first and then decreased as the NAA concentration increased. When the NAA concentration was held constant and the 6-BA concentration was 2.0 mg/L, the callus induction rate approached 82.3%, and browning rate was controlled within 6.7%. In addition, when the concentrations of 6-BA and NAA were 2.0 and 0.5 mg/L,

respectively, the callus induction rate of leaf discs in (1/2 N) MS medium, MS medium, and WPM medium were lower than that in 1/2 MS medium. By comparison, the most suitable medium formula for callus induction from the leaf discs of *D. lotus* was 1/2 MS medium supplemented with 2.0 mg/L 6-BA and 0.5 mg/L NAA.

**Table 3.** Effect of different plant growth regulators and media on callus induction in *D. lotus* leaf discs.

| Culture Medium | BA mg/L | NAA mg/L | Rate of Callus Induction % | Browning Rate % |
|---|---|---|---|---|
| 1/2 MS | 1.0 | 0 | 3.6 ± 1.6g (3) | 8.0 ± 1.3d |
| 1/2 MS | 2.0 | 0 | 15.2 ± 3.3f (14) | 7.3 ± 1.2d |
| 1/2 MS | 3.0 | 0 | 22.5 ± 3.6e (20) | 11.3 ± 2.3c |
| 1/2 MS | 1.0 | 0.5 | 55.4 ± 6.7c (50) | 6.2 ± 1.6d |
| 1/2 MS | 2.0 | 0.5 | 82.3 ± 3.7a (74) | 6.7 ± 1.7d |
| 1/2 MS | 3.0 | 0.5 | 67.4 ± 2.6b (61) | 11.5 ± 3.3c |
| 1/2 MS | 1.0 | 1.0 | 40.5 ± 3.3d (36) | 15.0 ± 4.4bc |
| 1/2 MS | 2.0 | 1.0 | 55.2 ± 2.6c (50) | 18.9 ± 1.3b |
| 1/2 MS | 3.0 | 1.0 | 43.7 ± 2.7d (39) | 25.3 ± 3.5a |
| (1/2N) MS | 2.0 | 0.5 | 72.5 ± 8.7b (65) | 8.7 ± 4.5d |
| MS | 2.0 | 0.5 | 80.5 ± 9.3a (72) | 19.7 ± 2.3b |
| WPM | 2.0 | 0.5 | 44.3 ± 8.6d (40) | 11.7 ± 3.3c |

Means followed by the same letters in rows are not significantly different at $p \leq 0.05$. The values in brackets are the real statistical data with the decimal point removed.

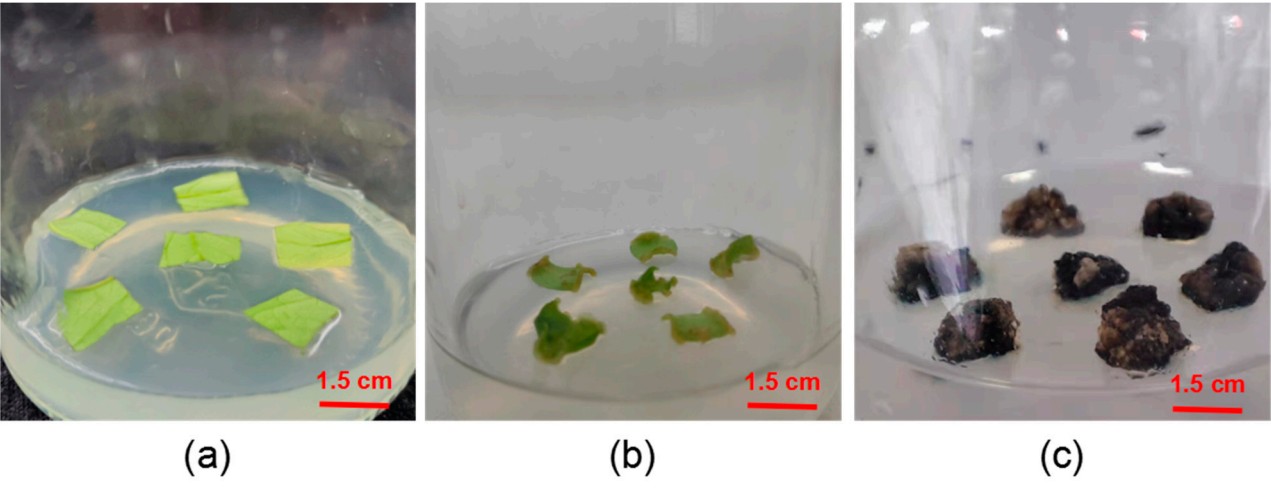

**Figure 2.** Process of leaves inducing calli. (**a**) Leaf discs connected to the culture medium after 5 days. (**b**) Callus leaf discs curled up after 10 days of culture. (**c**) Callus dimensions increase to $1.5 \times 1.5$ cm$^2$ after 25 days.

### 3.4. Induction of Adventitious Shoots from Calli

The induction of adventitious shoots is the most important step in leaf regeneration. Selecting a proper induction formula can effectively improve the induction efficiency and number of adventitious shots. From the results shown in Table 4 and Figure 3, the different concentrations of plant growth regulators showed significant differences in adventitious shoot induction of callus. In the process of callus-induced bud formation, the culture at 15 days exhibited a larger bud point (Figure 3a). Following bud formation, stem extension appeared after 5 days (Figure 3b). At 10 days, we observed leaf elongation accompanied by bud regeneration (Figure 3c). Single buds continued to grow taller and larger from 15 to 25 days (Figure 3d, e), with optimal bud growth observed at 30 days (Figure 3f). When the concentration of TDZ was 1.0 mg/L, the adventitious shoots induced by calli appeared yellow and had no obvious buds. At the same time, the overall multiplication coefficient and the rate of adventitious shoot induction were relatively low. When the TDZ concentration was 2.0 mg/L, the callus exhibited better overall growth, as well as large

leaves and strong stalks; the overall multiplication coefficient and the adventitious shoot induction rate were higher. At a TDZ concentration of 3.0 mg/L, the overall callus growth of adventitious shoots was very poor and the leaves were small and curled; the overall multiplication coefficient and the adventitious shoot induction rate were very low. The best adventitious shoot induction rate occurred in 2.0 mg/L TDZ, 2.0 mg/L 2iP, and 0.5 mg/L NAA cultures, in which the rate approached 75.2% and the average number of adventitious shoots reached 7.9; the growth was incredibly strong, and the leaves were big and green. By comparison, the most suitable medium formula for callus induction from the leaves of *D. lotus* was the 1/2 MS medium, supplemented with 2.0 mg/L 2iP, 2.0 mg/L TDZ, 0.5 mg/L NAA, and 40 g/L sucrose.

**Table 4.** Effects of different plant growth regulator concentrations, media, and sucrose concentration on the induction of adventitious shoots from calli of *D. lotus*.

| Culture Medium | 2iP mg/L | TDZ mg/L | NAA mg/L | Sucrose Concentration g/L | The Average Number of Adventitious Shoots | Adventitious Shoot Induction Rate (%) | Growth Condition |
|---|---|---|---|---|---|---|---|
| 1/2 MS | 0 | 0 | 0 | 0 | 0 | 0 | No adventitious shoots. |
| 1/2 MS | 0 | 1 | 0 | 40 | 0 | 0 | No adventitious shoots. |
| 1/2 MS | 0 | 2 | 0 | 40 | 1.2 ± 0.3f | 10.2 ± 2.7e | The leaves showed severe browning and the buds were prone to die. |
| 1/2 MS | 0 | 3 | 0 | 40 | 1.7 ± 0.5f | 14.5 ± 4.4e | The basal leaves showed severe browning. |
| 1/2 MS | 1 | 0 | 0 | 40 | 0 | 0 | No adventitious shoots. |
| 1/2 MS | 2 | 0 | 0 | 40 | 0 | 0 | No adventitious shoots. |
| 1/2 MS | 3 | 0 | 0 | 40 | 0 | 0 | No adventitious shoots. |
| 1/2 MS | 1 | 1 | 0.5 | 40 | 2.0 ± 1.3e | 30.4 ± 2.2c | The leaves were curled and small, and the buds were prone to die. |
| 1/2 MS | 2 | 1 | 0.5 | 40 | 2.3 ± 0.8e | 42.8 ± 3.1b | The leaves were large, and the adventitious shoots were prone to die. |
| 1/2 MS | 3 | 1 | 0.5 | 40 | 1.4 ± 0.3f | 36.7 ± 5.6c | The leaves were small and thin, with more cluster buds. |
| 1/2 MS | 1 | 2 | 0.5 | 40 | 4.2 ± 2.4c | 42.9 ± 7.3b | The leaves were smaller and grew more vigorously. |
| 1/2 MS | 2 | 2 | 0.5 | 40 | 7.9 ± 3.3a | 75.2 ± 8.4a | The leaves were large and green, and they grew extremely vigorously. |
| 1/2 MS | 3 | 2 | 0.5 | 40 | 5.5 ± 2.5b | 47.6 ± 3.3b | The leaves were fine, and the stem was tall and thick. |
| 1/2 MS | 1 | 3 | 0.5 | 40 | 2.2 ± 1.3e | 31.1 ± 3.4c | The basal leaves showed severe browning. |
| 1/2 MS | 2 | 3 | 0.5 | 40 | 4.2 ± 1.7c | 34.6 ± 9.2c | The leaves were yellow and the stems were short. |
| 1/2 MS | 3 | 3 | 0.5 | 40 | 1.1 ± 0.4f | 21.3 ± 1.1d | The leaves were curled up, and the stems were thick and long. |
| 1/2 MS | 2 | 2 | 0.5 | 30 | 4.1 ± 1.6c | 30.6 ± 5.4c | The basal leaves showed severe browning. |
| 1/2 MS | 2 | 2 | 0.5 | 50 | 4.4 ± 2.8c | 45.2 ± 6.7b | The leaves were yellow and small, and the base callus was severely browned. |
| (1/2N) MS | 2 | 2 | 0.5 | 40 | 3.9 ± 1.5d | 41.3 ± 5.8b | The leaves were relatively green, and the stems were slender. |
| MS | 2 | 2 | 0.5 | 40 | 2.7 ± 1.2e | 44.5 ± 7.6b | The leaves were pale green and small, and the stems were thick and long. |
| WPM | 2 | 2 | 0.5 | 40 | 2.9 ± 2.15e | 39.6 ± 4.1c | The leaves were large and green, and they grew extremely vigorously. |

Means followed by the same letters in rows are not significantly different at $p \leq 0.05$.

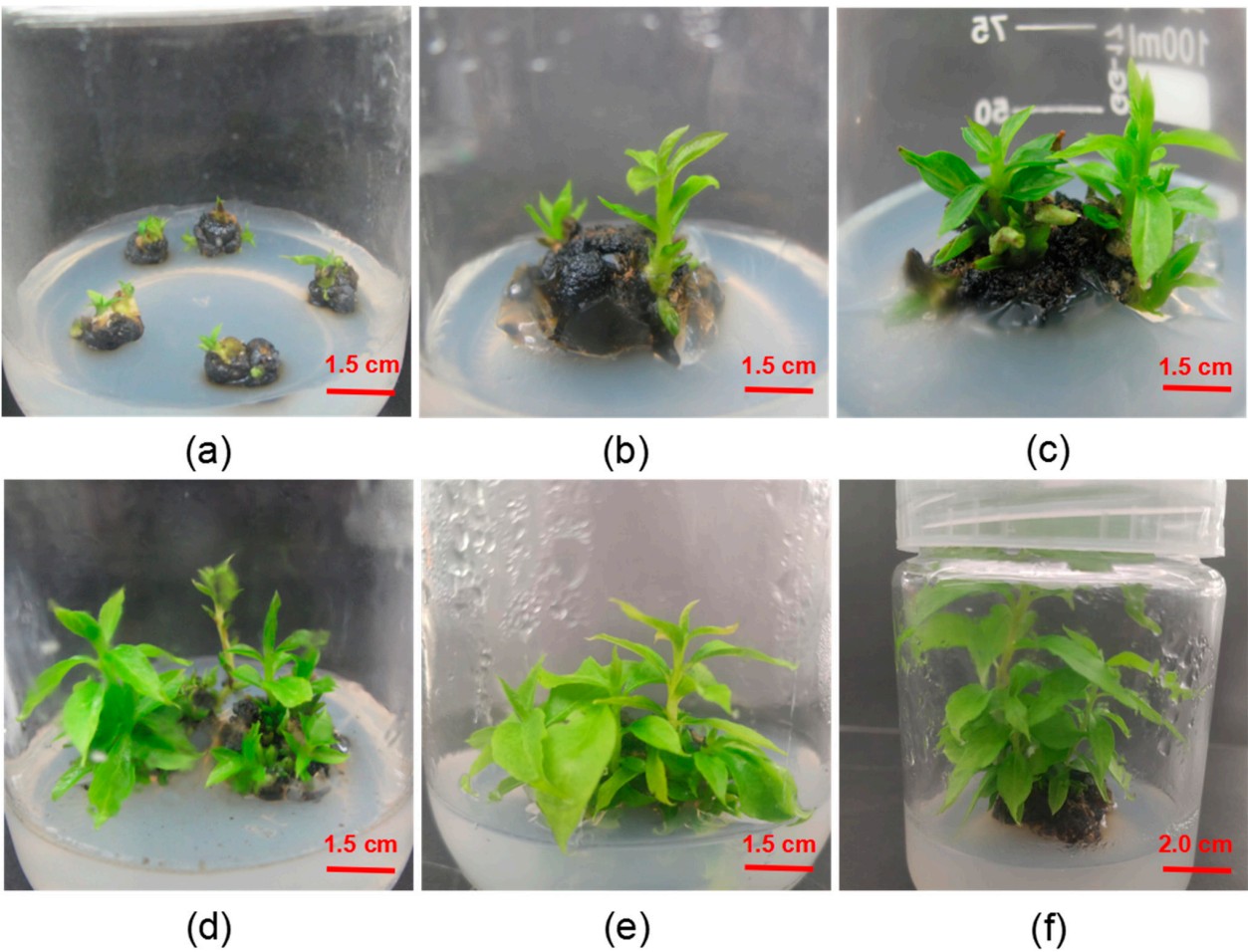

**Figure 3.** Callus and adventitious shoot induction from leaves in axillary buds of *D. lotus*. (**a**) Callus-induced adventitious shoots. Adventitious shoot growth after (**b**) 5 days, (**c**) 10 days, (**d**) 15 days, (**e**) 25 days, and (**f**) 30 days.

### 3.5. Growth of Calli after Multiple Induction of Adventitious Shoots

The callus often is brown and the adventitious shoots grow weakly, while the adventitious shoots induced by the callus are extremely weak with an increase in the number of adventitious shoots induced by the callus. The first-generation callus is the callus that forms after the first induction of adventitious shoots. From Table 5, the generations and induction rates of adventitious shoots were generally higher in the first, second, and third differentiation of callus formation, among which the induction rate of adventitious shoots was 80.0%, and the adventitious shoot coefficient was 5.3 in the second induction, with the best results. At the fourth differentiation, the value was 2.1 and the induction rate was 46.7%, specifically 30.7%, 33.3%, and 28.8% lower than that of the first, second, and third differentiations, respectively; the coefficients of adventitious shoots were 2.6, 3.2, and 3.0 lower, and the differences for both were significant ($P < 0.05$). At the fifth differentiation of adventitious shoots from callus, the growth of adventitious shoots was so weak that they could not be cultured in succession (Figure 4e). By comparison, the same callus showed optimal effects for the second and third differentiation buds.

**Table 5.** Callus-induced regeneration of adventitious shoots with respect to the bud number, bud induction rate, and growth condition.

| Generation of Differentiation | Adventitious Shoot Number | Unstable Bud Induction Rate % | Growth Condition |
|---|---|---|---|
| 1 | $4.7 \pm 1.3a$ | $77.8 \pm 4.1a$ (70) | The leaves were green but small, with smaller indefinite shoots. |
| 2 | $5.3 \pm 2.1a$ | $80.0 \pm 4.9a$ (72) | The leaves were large and green, and the adventitious shoots grew vigorously. |
| 3 | $5.1 \pm 2.3a$ | $75.5 \pm 11.1a$ (68) | The leaves were large, green, and robust. |
| 4 | $2.1 \pm 1.4b$ | $46.7 \pm 4.9c$ (42) | The leaves were dry and grew very weakly. |
| 5 | $1.0 \pm 0.4b$ | $31.1 \pm 2.49d$ (28) | The basal leaves showed severe browning. |

Means followed by the same letters in rows are not significantly different at $p \leq 0.05$. The values in brackets are the real statistical data with the decimal point removed.

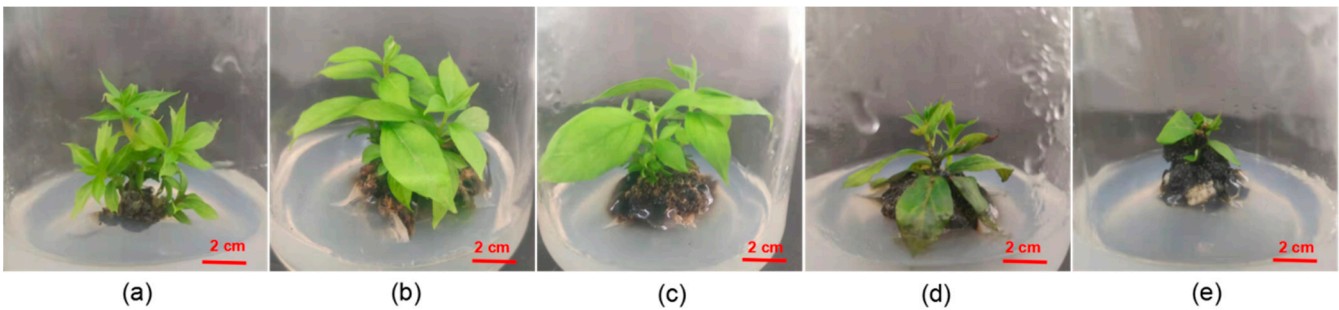

**Figure 4.** Growth of a callus after multiple inductions of adventitious shoots. Callus-induced adventitious shoot growth: (**a**) generation 1, (**b**) generation 2, (**c**) generation 3, (**d**) generation 4, and (**e**) generation 5.

### 3.6. Morphological Observation of Calli

In the process of subculture of calli it was found that the texture of the same callus would change greatly after the induction of adventitious shoots. Therefore, we observed the morphological changes of the bud point callus cells using sections. From Table 6, the fourth generation of callus cells was 18, 17, and 17 fewer in number than the first(Figure 5a), second(Figure 5b), and third(Figure 5c) generations under a $12\times$ microscopic field of view, respectively, with significant differences ($p < 0.05$). As the generation number increased from the fourth (Figure 5d) generation onward, the gap between the callus cells became larger, whereas the cells that were originally regularly distributed and similar in size became irregular and disorganized, especially the fifth generation (Figure 5e). Figure 5 a1–e1 shows the overall arrangement of the first to fifth generations of cells. From the fourth generation, the adventitious shoots of *D. lotus* callus grew weaker; thus, it was concluded that after three successive adventitious shoots from the same callus, the callus should be proliferated and cultured to induce robust adventitious shoots.

**Table 6.** Morphological data of callus cells in different generations.

| Proliferation Algebra | Cell Area µm² | Number of Cells | Cellular Morphology |
|---|---|---|---|
| First generation | $751.4 \pm 16.6c$ | $75 \pm 11a$ | The cell wall was intact, and the cells were arranged closely and regularly. The cells were mostly elliptical, and the cells near the bud point gathered inward and gradually became smaller. |
| Second generation | $738.9 \pm 25.2c$ | $74 \pm 10a$ | The cell wall was intact, and the cells showed a tight, regular arrangement. The lateral cells were large, and the cell wall in the area behind the bud point was completely broken. |
| Third generation | $766.6 \pm 31.5c$ | $74 \pm 12a$ | The cell wall was intact. The cell showed a close, regular arrangement. The lateral cells were long strips, and the internal test cells gradually became more relaxed. |

**Table 6.** *Cont.*

| Proliferation Algebra | Cell Area μm² | Number of Cells | Cellular Morphology |
|---|---|---|---|
| Fourth generation | 903.5 ± 43.7b | 57 ± 9b | The cell wall had ruptured, and the cells became loose and irregular. Most of the nuclei disappeared. |
| Fifth generation | 1266.3 ± 14.9a | 51 ± 6b | The cell wall had ruptured, the cells were loose and irregular, and the agglomeration in the callus had completely disappeared. |

Means followed by the same letters in rows are not significantly different at $p \leq 0.05$.

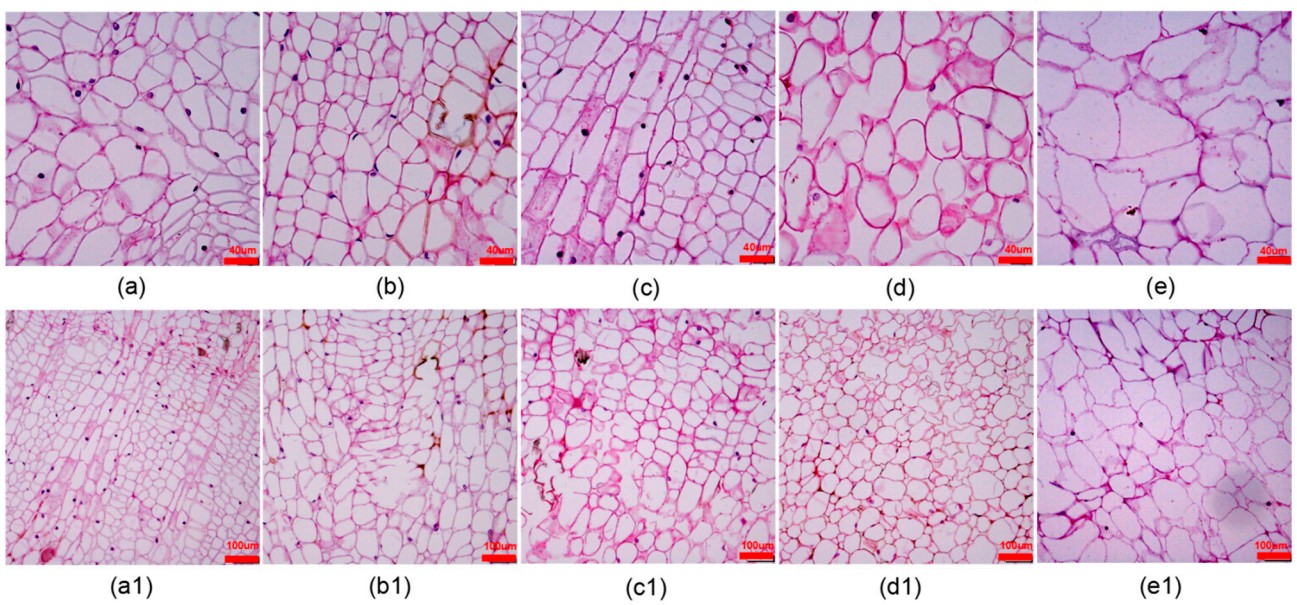

**Figure 5.** Callus cell morphology after repeated subcultures of callus of *D. lotus*. Generations 1–5 are shown in columns (**a**–**e**) (**a1**–**e1**), respectively.

*3.7. Adventitious Shoot Proliferation*

Subculture is an important way to obtain a large number of sterile saplings. Selecting a suitable culture method can effectively increase the multiplication coefficient and improve the growth of sterile saplings. Individual adventitious shoots were cut from the callus and inoculated into the culture medium as described in Table 7 (Figure 6a). When the callus contained no hormones, the indefinite stem grew slowly, and the multiplication coefficient was low. When the ZT concentration was held at a certain value, the multiplication coefficient and the average plant height increased at first after decreasing with the 2iP concentration. When ZT was 2.0 mg/L and IAA was 0.05 mg/L, the shoots showed strong growth at 10 days (Figure 6b). The growth after migration at 25 days and single-tufted stem growth after 15 and 30 days, respectively, are given in the remaining panels. When IAA was not added, the leaves in the medium were green. The culture medium most suitable for subsequent proliferation was 1/2 MS + 2.0 mg/L ZT + 2.0 mg/L 2iP + 0.05 mg/L IAA.

**Table 7.** Effects of different hormone combinations on the subculture and proliferation of *D. lotus*.

| IAA mg/L | 2iP mg/L | ZT mg/L | Multiplication Coefficient | Average Plant Height cm | Growth |
|---|---|---|---|---|---|
| 0.5 | 0 | 0 | 1.4 ± 0.3f | 1.3 ± 0.3d | Small leaves but long stems. |
| 0.05 | 1 | 1.5 | 3.5 ± 1.4e | 1.2 ± 0.2d | The leaves were long, yellow, and soft, and the stem sections were weak. |
| 0.05 | 2 | 1.5 | 5.5 ± 2.2c | 1.5 ± 0.1d | Leaves were small, but the overall growth was more prosperous. |
| 0.05 | 3 | 1.5 | 4.2 ± 2.3d | 1.4 ± 0.6d | The leaves were relatively thin, and some stem segments were weak and short. |
| 0.05 | 1 | 2 | 4.6 ± 1.7d | 2.6 ± 0.7b | The leaves were curled up, and the stems were soft but tall. |
| 0.05 | 2 | 2 | 7.6 ± 3.6a | 3.7 ± 0.5a | The leaves were green and large, and the stem segments were thick and tall. |
| 0.05 | 3 | 2 | 6.3 ± 2.5b | 2.9 ± 0.5b | The leaves were soft and yellow, and the leaves fell from the stems. |
| 0.05 | 1 | 2.5 | 4.5 ± 1.5d | 2.1 ± 0.6c | The leaves were smaller and more new shoots showed browning at the base. |
| 0.05 | 2 | 2.5 | 4.9 ± 1.2d | 2.3 ± 0.3c | Overall, the leaves were pale yellow with excessive base calli. |
| 0.05 | 3 | 2.5 | 2.7 ± 1.5f | 2.0 ± 0.5c | The leaves were small and soft, and the stem segments were short and weak. |
| 0 | 2 | 2 | 5.9 ± 2.1b | 1.5 ± 0.3d | The leaves were relatively green, with short and soft stems. |
| 0 | 1 | 0 | — | 1.2± 0.1d | No proliferation buds appeared. |
| 0 | 2 | 0 | — | 1.1± 0.2d | No proliferation buds appeared. |
| 0 | 3 | 0 | — | 1.3 ± 0.4d | No proliferation buds appeared. |
| 0 | 0 | 1 | — | 1.2 ± 0.1d | No proliferation buds appeared. |
| 0 | 0 | 2 | 1.1 ± 0.5f | 1.5 ± 0.2d | The leaves were yellow and a large number of calli were produced at the base. |
| 0 | 0 | 3 | 1.3 ± 0.5f | 1.5 ± 0.3d | The stems were short and a large number of calli were produced at the base. |

Means followed by the same letters in rows are not significantly different at $p \leq 0.05$.

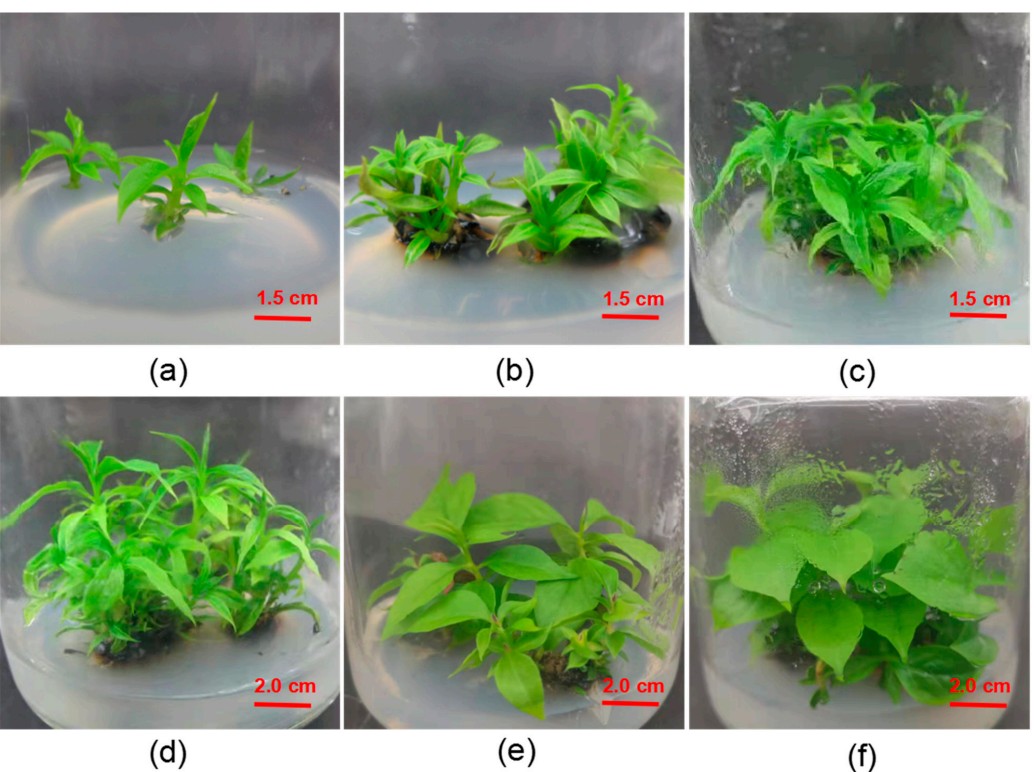

**Figure 6.** *D. lotus* subsequent proliferation culture. (**a**) Single adventitious shoots were incorporated into the medium. (**b**) Clump stems grew after 10 days. (**c**, **d**) Growth status of migration after 25 days. (**e**) Single-tufted stems cultured for 15 days. (**f**) Single-tufted stems cultured for 30 days.

*3.8. Rooting Culture*

Rooting culture is an important step in factory sapling cultivation. Selecting a suitable medium can effectively improve the effect of rooting culture and increase the survival rate of saplings. Individual adventitious shoots were cut from the tufted stems and inoculated into the culture medium as described in Table 8 (Figure 7a). When the IBA concentration was 0.5 mg/L, the rooting rate and rooting number were lower as a whole; when the concentration of IBA was 1.0 mg/L, the rooting rate increased first and then decreased with the concentration of NAA. When the concentration of NAA was 0.5 mg/L, the rooting rate was relatively high; when the IBA concentration was 1.5 mg/L, the rooting rate and rooting number were lower as a whole. The study on the effect of IBA and NAA on rooting showed that when IBA was 1.0 mg/L and NAA was 0.5 mg/L, the rooting rate was higher as a whole; when the IBA was 1.0 mg/L and the NAA was 0.5 mg / L, with the increase in sucrose concentration, the rooting rate and rooting number first increase and then decrease. When the sucrose concentration was 20 g/L, the rooting rate and rooting number were the highest, and the growth was excellent. After 25 days of rooting culture, a large number of adventitious roots (Figure 7b) began to be produced. After 40 days of culture, the leaves were large and green (Figure 7c), the roots grew vigorously (Figure 7d), and the adventitious shoots of rooting culture grew very well overall (Figure 7e). Finally, the best culture and combination of root culture was 1/2 MS + 1.0 mg/L IBA + 0.5 mg/L NAA + 20 g/L sucrose.

**Table 8.** Effects of different basic media and hormones on rooting culture of *D. lotus*.

| IBA mg/L | NAA mg/L | Sucrose Concentration g/L | Take Root Rate % | Adventitious Roots Number |
|---|---|---|---|---|
| 0 | 0 | 0 | 0 | 0 |
| 0.5 | 0 | 15 | 15.3 ± 1.3d (14) | 2.3 ± 1.5d |
| 0.5 | 0.5 | 15 | 32.1 ± 1.4c (29) | 3.2 ± 1.7c |
| 0.5 | 1.0 | 15 | 28.4 ± 4.1c (26) | 2.5 ± 1.4e |
| 1.0 | 0 | 15 | 24.6 ± 3.1c (22) | 4.2 ± 0.2b |
| 1.0 | 0.5 | 15 | 40.1 ± 2.6b (36) | 3.4 ± 1.2c |
| 1.0 | 1.0 | 15 | 23.2 ± 4.7c (21) | 2.2 ± 0.2d |
| 1.5 | 0 | 15 | 22.1 ± 1.3c (20) | 1.5 ± 0.7e |
| 1.5 | 0.5 | 15 | 40.4 ± 3.5b (36) | 1.5 ± 1.1e |
| 1.5 | 1.0 | 15 | 21.0 ± 2.4c (19) | 4.7 ± 2.3b |
| 1.0 | 0.5 | 20 | 70.2 ± 2.2a (63) | 9.6 ± 2.3a |
| 1.0 | 0.5 | 25 | 50.1 ± 2.6b (45) | 3.4 ± 1.2c |
| 0 | 0.5 | 20 | 0 | 0 |
| 0 | 1.0 | 20 | 0 | 0 |

Means followed by the same letters in rows are not significantly different at $p \leq 0.05$. The values in brackets are the real statistical data with the decimal point removed.

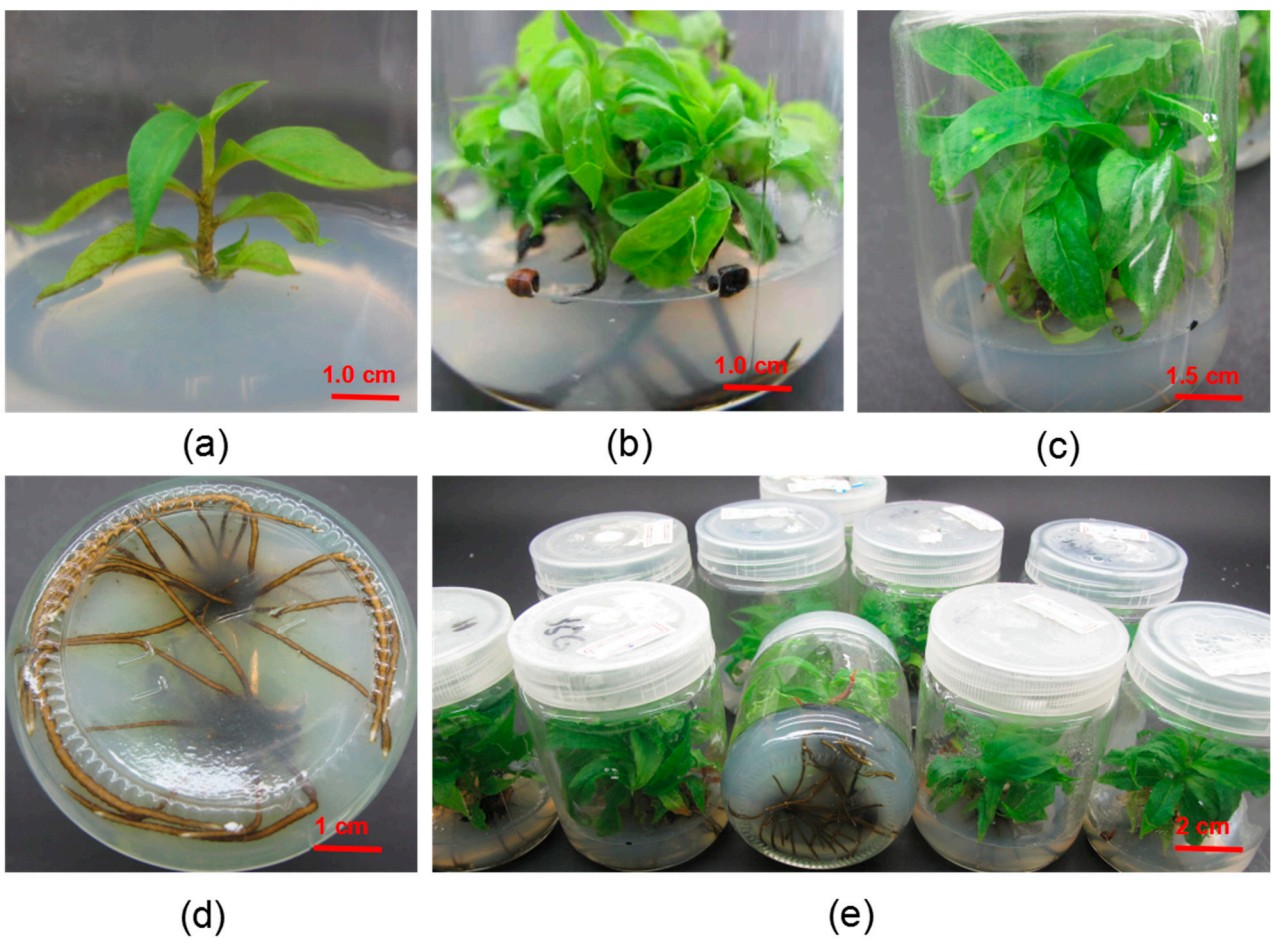

**Figure 7.** Rooting Culture of *D. lotus* (**a**) The stem segments of single adventitious shoots were lignified after 10 days of culture. (**b**) Adventitious roots appeared after 25 days of rooting culture. (**c**) The growth of the upper part of root after 40 days of rooting culture. (**d**) Root growth after 40 days of rooting culture. (**e**) Overall photo of rooting culture.

## 4. Discussion

Disinfection of explants is the most critical part of plant tissue culture, given the rapid propagation of woody plant tissue [12]. In addition, with the growth and improvement of lignification of explants, the disinfection time must be extended [13]. Persimmon contains a high level of tannins and phenolic substances. Phenolic substances can cause browning of their explants during tissue culture, which can seriously affect the normal growth of histoculture saplings [14]. This study focused on this issue in the early stages of *D. lotus* tissue culture, in an effort to minimize browning effects. In the study of the primary culture of the stem with a bud segment of *D. lotus*, it was found that different disinfectants, the disinfection time, and the sequence of treatment all had significant effects on the growth of histoponic saplings. Stem segments with buds of a 3-year-old sapling were used as explants, which not only effectively solved the browning problem of *D. lotus* but also significantly improved the survival rate of *D. lotus* stems with buds. Most of the stems with buds of *D. lotus* studied by predecessors were dormant buds that were cultured after sterilization with anti-browning agents [11]. By contrast, the use of sprouted stem segments as explants not only allowed for culture without adding anti-browning agents after sterilization but also improved the culture efficiency. In vitro tissue culture of woody plants frequently uses the cytokinins TDZ, 2iP, and ZT, as well as the growth factors NAA, indole-3-butyric acid (IBA), and IAA. Currently, the most commonly used cytokinin in persimmon tissue culture is ZT, which plays a key role in the proliferation and growth of the saplings; moreover, persimmons have a specific dependence on ZT during the proliferation culture [15–17].

In addition, the effect and efficiency of *D. lotus* succession culture could be improved by adding 2.0 mg/L 2iP. The average multiplication coefficient of *D. lotus* approached 7.6 when cultured in 1/2 MS + 2.0 mg/L ZT + 0.05 mg/L IAA + 2.0 mg/L 2iP medium. The plant growth regulator plays a role in axillary growth in the tissue culture of woody plants. In the process of *D. lotus* proliferation and culture, when IAA was not added, the leaves were small and yellow, which seriously affected the photosynthesis of the plant. The proliferation efficiency was low and the cluster buds could not grow. The leaves fell, and the shoots were yellow after 30 days of cultivation. After adding IAA, the cluster shoot growth was extremely strong; 20 days of culture produced a dense shoot and long, thick shoots. This is consistent with the results of Chang et al. (2016) in *Diospyros* kaki (cv. Youhou) [18].

In the persimmon tissue culture process, NAA is commonly used in the regeneration step. In this study, NAA played an important role in the process of adventitious shoot induction of callus from leaf disc. The addition of NAA effectively promotes cell division and expansion, thus facilitating rapid callus formation. In the process of callus induction, NAA induces shoot leaf and cell expansion for improved shoot growth and a shorter regeneration time, as observed by Liu et al. (2007) in *Diospyros kaki* Thunb [19]. Cytokinins promote rapid cell differentiation for tissue growth and regeneration. TDZ facilitates callus shoot induction. TDZ in a concentration of 2.0 mg/L accelerated shoot regeneration and had a greater effect than 6-BA. Notably, high TDZ concentrations led to very weak callus-induced shoot growth, similar to the results of Nauli et al. in wild peony [20] and Li et al. (2018) in *D. lotus* [5]. In regeneration systems, TDZ is relatively stable. Thus, long-term storage solutions containing TDZ can cause physiological reactions. Recent studies have found that TDZ molecules form short and long multimers during extended storage periods, and that this polymerization is found in both the solution and the medium, thereby producing adventitious shoots via callus stimulation [20].

In the *D. lotus* proliferation culture, calli induced indefinite shoot formation via 2iP. Without 2iP or with low 2iP concentrations, the leaves tended to yellow and the growth rate slowed. High 2iP concentrations increased the multiplication coefficient but also resulted in insufficient nutrition, resulting in very weak shoots. In the process of a callus-induced variable shoot, this problem also exists, i.e., high 2iP concentrations promote indefinite formation but the shoots are weak. This is likely due to the ability of 2iP to stimulate plant cell division and chlorophyll formation [21]. In the process of tissue culture, there is a phenomenon of small and soft leaves. 2iP addition can improve this, such that the leaf area of *D. lotus* is enhanced, which is conducive to photosynthesis and plant adaptability. In the process of persimmon root cultivation, a previous study demonstrated the IBA-induced root effect in persimmon group saplings, and the transplant survival rate was high [22]. The difficulty of plant rooting is related to the differences among varieties but strongly depends on the endogenous hormone level of the plant itself [23]. In addition, the type and concentration of hormones added to the medium and the choice of rooting method can affect the rooting of plants to different degrees. Hormones commonly used in rooting culture include IBA, NAA, and IAA [24]. IBA is the most commonly used growth factor in the rooting of plant saplings in group cultures to promote primary root growth, whereas NAA induces cell division and expansion and the formation of adventitious roots. In this study, the combination of IBA and NAA was chosen to induce adventitious roots in the rooting culture, with good results.

## 5. Conclusions

The stem segments with buds were sterilized with 75% ethanol for 25 s, then sterilized with 0.1% $HgCl_2$ for 25 min, and then inoculated into 1/2 MS + 2.0 mg/L 6-BA + 0.5 mg/L NAA medium. The axillary bud sprouting rate was 67.1%, the pollution rate was 11.7%, and the average sprouting time was 17.2 days. The medium formula for axillary bud subculture was 1/2 MS + 2.0 mg/L ZT + 2.0 mg/L 2ip + 0.05 mg/L IAA. The medium formula for callus induction from *D. lotus* leaves was 1/2 MS medium supplemented with 2.0 mg/L 6-BA and 0.5 mg/L NAA. The best medium for inducing adventitious shoots from calli was 1/2 MS

+ 2.0 mg/L TDZ + 0.5 mg/L NAA + 2.0 mg/L 2ip + 40 g/L sucrose, the induction rate of adventitious shoots was 75.2%, and the average number of adventitious shoots was 7.9 after 25 days of culture. The optimum medium for the rooting culture of *D. lotus* was 1/2 MS + 1.0 mg/L IBA + 0.5 mg/L NAA. The average number of roots and the rooting rate reached 9.6 and 70.2%, respectively, after being cultured for 40 days and in the dark for 5 days.

## 6. Patents

There is a patent that resulted from the work reported, which is now in the stage of publication and substantive review. Ze Li, Yang Liu, Hui Zhang, Jian Min Fu, Shu Zhan Li, Peng Sun, Ting Zhang, Zi Yan Xu, A method for obtaining regenerated plants from the shoot segment with buds of *Diospyros lotus*, 2022.11.7, China, 202,210,762,267.9.

**Author Contributions:** Conceptualization, Y.L. and Z.L.; methodology, Y.L. and Z.L.; validation, H.Z. and X.L.; formal analysis, Y.L. and H.Z.; investigation, X.L. and Y.L.; resources, Y.L.; data curation, Y.L.; writing—original draft, Y.L.; writing—review and editing, Z.L.; visualization, Y.L.; supervision, Z.L. and S.L.; project administration, Z.L.; data curation, Z.L. All authors have read and agreed to the published version of the manuscript.

**Funding:** This study was supported by the National Key R&D Program of China, project number is 2018YFD1000606, project leader is Ze Li.

**Data Availability Statement:** All data are available in the manuscript.

**Conflicts of Interest:** The authors declare no conflict of interest.

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
