# Peer review of "Establishment of a Highly Efficient In Vitro Propagation System of Diospyros lotus"

_forests, doi:10.3390/f14020366_

Round 1

Reviewer 1 Report (New Reviewer)

The authors have conducted a detailed study of the conditions for tissue culture of Diospyros lotus, particularly with regard to asepsis, which is said to be highly difficult in woody plants, and the composition of the medium, mainly phytohormones, during induction of callus, adventitious shoots, and roots. The experiments are meticulously planned and the conclusions drawn from the data are clear. Although there is no novelty in the successful tissue culture or transformation of Diospyros lotus, this is a valuable study in terms of accumulation of knowledge.

Although I do not see any major problems with the structure of the paper or the English language, I have some concerns about the following points, which I would appreciate an answer or response.

(1) In the results section, the purpose and necessity of each experiment are not described, and I have the strong impression that the results are just a list of results. I would like you to add an explanation for each item, even if it is simple, and create a flow to the discussion.

(2) The evidence that the cell-level differences shown in Fig. 5 are linked to the indeterminate bud re-differentiation rate is insufficient, and I feel that the evidence is not beyond the scope of estimation. If there is any previous knowledge on the relationship between the two, please describe it in more detail and add where the sections used for microscopic observation were taken from in the callus.

(3) As for the results of the tissue culture experiment, I have the impression that there is too little variation in the data for the number of n, and I feel a strong sense of discomfort (the data is too beautiful). I would like you to show the actual numbers in each table, not just the calculated values.

Author Response

Response to Reviewer 1 Comments

Point 1: In the results section, the purpose and necessity of each experiment are not described, and I have the strong impression that the results are just a list of results. I would like you to add an explanation for each item, even if it is simple, and create a flow to the discussion.

Response 1: Thank you for your suggestion. We have solved this question and added the necessity of relevant research in front of each result. The main purpose is to ensure that readers of this article can find the key points of research at each stage while watching the results, and know in advance the cultivation methods needed in the later genetic transformation research, so as to provide new cultivation ideas for persimmon genetic transformation. Specific additions are listed in line 159,177,204,226,261,284,301 and 323.

Point 2: The evidence that the cell-level differences shown in Fig. 5 are linked to the indeterminate bud re-differentiation rate is insufficient, and I feel that the evidence is not beyond the scope of estimation. If there is any previous knowledge on the relationship between the two, please describe it in more detail and add where the sections used for microscopic observation were taken from in the callus.

Response 2: Thank you for your suggestion. We have added the location of sliced callus in the manuscript. In this study, the selected position of the callus is the position where the adventitious buds grow. We selected 150 bottles of the callus with the same growth. After the adventitious buds grow from the callus, we will cut the upper adventitious buds and cut from the lower callus 5 ×5 × 5 mm3 in size and stored in FAA buffer solution, put the remaining calli into the culture medium to continue to induce calli. During this process, continue to take photos and observe the growth of adventitious buds. The calli used are all the same piece. After the sixth time, we found that the calli could not induce adventitious buds, and the whole interior was browning when cut. Because we found that the adventitious buds could be induced continuously by inserting the original medium after the adventitious buds were induced by the callus, which greatly shortened the seedling raising time and culture cost, we conducted this study, and finally found that the same callus could effectively reduce the seedling raising cost by subculturing the callus after three consecutive adventitious buds were induced in the same culture medium, It provides a new idea for the difficulty of breeding of positive seedlings after genetic transformation.

Point 3: As for the results of the tissue culture experiment, I have the impression that there is too little variation in the data for the number of n, and I feel a strong sense of discomfort (the data is too beautiful). I would like you to show the actual numbers in each table, not just the calculated values.

Response 3: Thank you for your suggestion. For some data with relatively small statistical quantity, we expanded the data volume to 90, that is, the combination of three groups of data, and made statistical analysis of this data. We have added specific data in the article. This part of data is the average value of decimals, which is indicated in brackets. For the data with large changes, we have changed all the data. No changes have been made to the parts with clear figures such as multiplication coefficient, growth days and rooting number. I hope that such changes can get your approval.

Reviewer 2 Report (New Reviewer)

To the authors/editor,

The manuscript entitled „Establishment of a highly efficient regeneration system of Diospyros lotus” presents the results of a detailed and extensive experiment resulting in a clear protocol for the effective multiplication of plant material of D. lotus.

I suggest a slight change in the title of the manuscript: Establishment of a highly efficient in vitro propagation system of Diospyros lotus” .

The significance of the topic, including rootstocks for D.kaki  production, rootstock breeding issues and tissue culture issues are sufficiently elaborated in the introduction section.

Materials and methods are clearly and sufficiently presented enabling the repeating of the experiment. The only issue related to methodology is concerning the statistical analysis. I suggest authors to consider the possibility of performing factorial ANOVA and to present the concentrations of single substances in growing medium as experimental factors as well as their interactions.

Tables and figures/photos are sufficient and clearly present the data. However, the main flaw of the manuscript is related to terminology issues which were noticed in results and discussion sections and given in specific comments and in the manuscript text comments.

In certain points, authors use “seedlings” for plant material which is not derived from seeds. I suggest checking the use of the term “seedling” in whole text and using “saplings” instead, or other appropriate term where needed (given in text comments lines 131, 339, 353, 401 etc).

Line 158. The terms used in text should be in line with terms used in results presentation in tables and figures. At this point, authors use and discuss on “survival rate” in the text, but not in the table 1. Need to be clarified.

Line 172. After sterilization experiment, which sterilization procedure was used for sterilization of plant material for further experiment?

Line 175. Seems that many times in text authors misused word “bud” instead using “stem”. I suggest authors to check throughout the whole text very carefully, for example lines 218, 224, 226, header in table 4, 260, 265 etc.

Line 177. Seems like term “germination rate” is misused. Germination is commonly a term related to growing from seeds which is not the present case. In table 2, term “sprouting is used”. Please make this celar.

Line 199, 201, table 3. Is it rate of callus formation or rate of callus induction

Line 223. What is proliferation coefficient? There is no such in the table 4.

Line 293, table 7. Is it proliferation coefficient or multiplication coefficient?

Above listed and other (given in the text comments) terminology issues must be clarified.

 Sincerely,

Reviewer

Author Response

Response to Reviewer 2 Comments

Point 1: I suggest a slight change in the title of the manuscript: “Establishment of a highly efficient in vitro propagation system of Diospyros lotus.”

Response 1: We have changed the title to “Establishment of a highly effective in the promotion system of Diospyros lotus” according to your suggestion. (line 2)

Point 2: Materials and methods are clearly and sufficiently presented enabling the repeating of the experiment. The only issue related to methodology is concerning the statistical analysis. I suggest authors to consider the possibility of performing factorial ANOVA and to present the concentrations of single substances in growing medium as experimental factors as well as their interactions.

Response 2: We have added the concentration of a single substance as the relevant data of experimental factors in the table. However, in the process of callus induction of adventitious buds (3.4), because a single auxin could not induce adventitious buds from the callus, we only added the effect of a single cytokinin on callus induction of adventitious buds (Table 4,7and 8). I hope our modification can get your approval.

Point 3: In certain points, authors use “seedlings” for plant material which is not derived from seeds. I suggest checking the use of the term “seedling” in whole text and using “saplings” instead, or other appropriate term where needed (given in text comments lines 131, 339, 353, 401 etc).

Response 3: We are very sorry for our incorrect writing seedlings. We have changed “seedling” instead of “saplings” in the manuscript.

Point 4: Line 158. The terms used in text should be in line with terms used in results presentation in tables and figures. At this point, authors use and discuss on “survival rate” in the text, but not in the table 1. Need to be clarified.

Response 4: Thank you very much for your question. Due to my own negligence, the title corresponding to the data in the table is not consistent with the text. Now the terms in 158 lines have been unified.

Point 5: Line 172. After sterilization experiment, which sterilization procedure was used for sterilization of plant material for further experiment?

Response 5: After the sterilization test, the method we used was using 75% ethanol for 30 s, followed by 0.1% HgCl2 for 25 min. This step was added at the beginning of the subsequent test results.

Point 6: Line 175. Seems that many times in text authors misused word “bud” instead using “stem”. I suggest authors to check throughout the whole text very carefully, for example lines 218, 224, 226, header in table 4, 260, 265 etc.

Response 6: Thank you very much for your careful review. Thank you very much for your question. our statistical data is the number of adventitious shoots induced by callus, The main reason for using 'shoot' in this part is fixed collocation, while stem mainly describes the lower part of flowers and plants, so the bud in this part has not been modified. So we use shoot to describe bud and stem.

Point 7: Line 177. Seems like term “germination rate” is misused. Germination is commonly a term related to growing from seeds which is not the present case. In table 2, term“sprouting is used”. Please make this clear.

Response 7: We change the germination rate to sprouting rate, which is more suitable for the description of axillary bud germination in the stem segment with buds. In addition, we also reviewed relevant literature and finally decided to use "sprouting".

Point 8: Line 199, 201, table 3. Is it rate of callus formation or rate of callus induction

Response 8: Thank you very much for your question. We are very sorry for our incorrect writing rate of callus formation. It should be rate of call induction. We have revised it in the manuscript according to your suggestion.

Point 9: Line 223. What is proliferation coefficient? There is no such in the table 4.

Response 9: Thank you very much for your meticulous work and rigorous academic attitude. In Table 4, it should be the average number of adventitious shoots. We have revised it in the manuscript according to your suggestion.

Point 10: Line 293, table 7. Is it proliferation coefficient or multiplication coefficient?

Response 10: It is multiple coefficient. We have changed the title to “multiple coefficient” in the manuscript according to your suggestion.

This manuscript is a resubmission of an earlier submission. The following is a list of the peer review reports and author responses from that submission.

Round 1

Reviewer 1 Report

Dear PhD Li Ze, first of all let me congratulate you and to your co-author for this great research related to the possible establishment of a cell line isolated from a species with great application in the timber sector, being a feasible possibility that allows the best use of the species through cell cultures of high regeneration in its bark and quick growth to the development of fruits, without generating negative impacts on the environment. The following comments will only be for the purpose of further improving the quality and format of your manuscript, so that your results are better understood by readers:

ABSTRACT

-          Just please specify in the first part what is the main use of the species, and that the objective of the work is to improve the grafting process by in vitro culture of this species.

-          Please mention in the part of the methodology where and how the samples were taken to be able to perform the cultures in the MS medium.

-          Please, in the part of the results were missing mention important parameters such as those of growth kinetics, add them, as well as the total cultivation time and sub-culture cycles.

INTRODUCTION

-          Please refocus the background of the species more than towards economic use in the timber industry, than towards ethnomedicinal uses, however if more importance will be given to the fruits rewrite in the abstract focusing on the search to create more fruits managing shorter growth times and maturity of new shoots

-          As a final paragraph of this section please put the objective of this work and its importance or innovation.

MATERIALS AND METHODS

-          As first sub-section please add to the list of all the chemical reagents, test kits, and measurement instruments used in the experimental development information related to catalog number and company from where were bought, and for measurement instruments: model, company, and specifications.

-          A part must be added where the taxonomic identification of the species by a botanist and the elaboration of herbarium specimen is mentioned and mention the voucher number.

-          Take care of the correct use of capitalization in subtitles as in number 2.4.

-          In line 103 correct “Caucus” for “Callus”.

-          Please mention how the results of each technique were reported. Also, all these such must be referenced to previous published work.

-          Please mention the number of explants as (n = 30) for each of the techniques that require it.

-          Please change in line 136, "P" to "p" in italics, make this correction in all text where applicable.

-          Please reposition where best appropriate the formulas of the calculations described in lines 139 to 147.

RESULTS

-          Please put as additional information in all table footers the number of specimens that received the treatments, for example for Table 1, in line 163, add (n = 25). Add this data in all tables where appropriate.

-          Please put the figures or tables immediately after the text quoting or referring to them.

-          In the figure caption of the photos of the tissues analyzed, the objective or zoom that was used in the microscope for the observation should be added, also mention the method used to stain.

-          Please, in the part of the results were missing mention important parameters such as those of growth kinetics, add them, as well as the total cultivation time and sub-culture cycles, as well their mention in the methods section.

DISCUSSIONS

-          No comments.

CONCLUSIONS

-          A final paragraph is required after the discussions, where the most outstanding results of the study are described in a concrete and forceful way, as well as its perspectives.

Congratulations is a great experimental work, continue with the good work.

Reviewer 2 Report

Dear authors,

There were many approaches with establishment of efficient genetic transformation systems, such as Scientia Horticulturae (Li et al. 2018), Horticulturae (Zhang et al., 2022), Regulation of Plant Growth & Development (Tao et al. 2001), and one recently published from your group Scientia Horticulturae (Ma et al., 2023). In these manuscript, they have already established efficient regeneration systems in Diospyros.

More specially, in WOS database, these research groups like profs. Jihong Liu and Zhengrong Luo (HZAU), Donghong Zhuang (Shantou University), and more previously from Ryuturo Tao (Kyoto University), have been already established very very similar approaches to produce regeneration system in D. lotus.

Thus, I would suggest the authors to provide the establishment of novel efficient transgenic systems, like CRISPR-Cas9 tool? Please kindly note that, even for an efficient Agrobacterium-mediated genetic transformation systems that have already been established with many times in Diospyros species, these trangenic lines with a GFP or GUS reporter gene is needed to confirm the regeneration hygromycin resistant plantlets.

Or authors can conduct a successful transgenic example? For instance, overexpression of FT will introduce early flowering in D. lotus?

I am sorry for that I do not have more positive comments with current version of this mansucript, because when I search articles related to this field, there are lots of publications.